# Effect of Exercise Using an Exoskeletal Hip-Assist Robot on Physical Function and Walking Efficiency in Older Adults

**DOI:** 10.3390/jpm12122077

**Published:** 2022-12-16

**Authors:** Su-Hyun Lee, Hwang-Jae Lee, Kyungrock Kim, Byoung-Hee Lee, Yun-Hee Kim

**Affiliations:** 1Center for Prevention and Rehabilitation, Samsung Medical Center, Department of Physical and Rehabilitation Medicine, Heart Vascular Stroke Institute, Sungkyunkwan University School of Medicine, Seoul 06351, Republic of Korea; 2Robot Business Team, Samsung Electronics, Suwon 16499, Republic of Korea; 3GEMS Lab, Samsung Research, Samsung Electronics, Seoul 06765, Republic of Korea; 4Department of Physical Therapy, Sahmyook University, Seoul 01795, Republic of Korea; 5Department of Health Sciences and Technology, SAIHST, Sungkyunkwan University, Seoul 06351, Republic of Korea; 6Department of Medical Device Management and Research, SAIHST, Sungkyunkwan University, Seoul 06351, Republic of Korea; 7Department of Digital Health, SAIHST, Sungkyunkwan University, Seoul 06351, Republic of Korea

**Keywords:** older adults, physical function, walking efficiency, wearable hip exoskeleton

## Abstract

Robotic technology has developed rapidly in recent years, and several robotic devices have been applied to improve physical, sensory, intellectual, psychological, and social functioning in the elderly and people with disabilities. In this study, we investigated the effects of EX1-assisted exercise in various environments on physical function, muscle strength, and walking efficiency in older adults. We designated four experimental conditions and randomly assigned participants to one of four groups: A (overground walking without an EX1), B (overground walking using the resistance mode of EX1), C (stair ascent using the assistance mode of EX1), and Group D (inclined treadmill walking using the assistance mode of EX1). A total of 60 community-dwelling elderly persons participated in 10 exercise intervention sessions for 4 weeks, and all participants were assessed before and after the exercise intervention. Physical function was measured by the 10-meter walk test for self-selected velocity (10MWT-SSV), short physical performance battery (SPPB), Berg balance scale (BBS), timed up and go (TUG), functional reach test (FRT), geriatric depression scale—short form (GDS-SF), and muscle strength of trunk and lower extremity. Cardiopulmonary metabolic energy efficiency was measured using a portable telemetric gas analyzer system. A significant increase in the 10MWT-SSV and TUG test was observed in groups B, C, and D. A statistically significant improvement in the SPPB and FRT was seen only in group D, and GDS-SF scores decreased significantly after exercise with an EX1 in groups B and D. Trunk and lower limb muscle strength increased more in the groups that exercised with EX1 assistance than those without an EX1, particularly in group B. The net metabolic energy costs and energy expenditure measurement during walking significantly improved in exercise groups C and D. The findings in this study support the application of the EX1 to physical activity and exercise to improve age-related changes in physical function, muscle strength, and walking efficiency among older adults. In addition, personalized exercise programs using different modes and training environments with an EX1 can enhance physical performance and walking efficiency in the elderly.

## 1. Introduction

Older adults generally experience decreases in their physical abilities due to deterioration of the neuromusculoskeletal system, which leads to a changed, more cautious gait [1,2]. Increased age is associated with a decrease in preferred gait speed, stride length, and range of motion of the ankle, knee, and hip joints and an increase in step variability, stance width, time spent in the double support phase (i.e., with both feet on the ground), and metabolic cost of walking [3,4,5]. These age-related changes may represent adaptations to changes in sensory or motor systems that produce a safer and more stable gait pattern [6]. In many cases, a decline in physical and gait function ultimately leads to a sedentary lifestyle, which is strongly correlated with a variety of metabolic and cardiovascular diseases [7]. Life expectancy is increasing in most countries, and, as a result, rapid population aging is being reported worldwide. Because the prevalence of gait and physical disorders increases with age, the number of people affected by these disorders will increase significantly in the coming decades [1].

Physical activity and exercise are recommended strategies to mitigate age-related declines in gait and physical function in older adults [8,9]. Participation in physical activities and exercise can help with improving or maintaining physical function and overall health and reducing falls among elderly adults, particularly those with morbidities [10,11]. Physical activity and exercise are also associated with improved mental health, delayed onset of dementia, and improved quality of life and well-being [12,13]. Additionally, a lack of exercise leads to an increased incidence of chronic diseases, which increases medical expenses and the economic burden of individuals and the state. Compared with the high cost of drugs, exercise intervention is an economic and safe way to prevent and treat diseases and has few side effects; this strategy thus reduces the economic burden of chronic diseases on families and society. Therefore, guiding people to take part in exercise to enhance physical fitness is more important than ever [14].

Aerobic exercise is a well-established approach to improving aerobic capacity and health. Aerobic exercise has many cardiovascular and musculoskeletal benefits for elderly people [15,16] and plays a homeostatic role in regulating the rate of energy production, blood flow, and substrate utilization in response to locomotion [17]. Resistance exercise is recommended to help adults maintain their overall health and improves muscle mass, strength, power, and quality, as well as overall physical function in older adults [16,18,19]. These exercise programs can improve mobility, physical function, and performance of daily activities; enhance psychosocial well-being; and preserve the independence of older adults [20]. Multicomponent exercise (combined aerobic and resistance training) has a more comprehensive and practical effect on physical performance among the elderly [21,22,23]. Previous theoretical studies showed that combining aerobic with resistance training not only enhances cardiorespiratory function but also increases muscle strength [24,25].

Stair climbing is one of the most challenging and essential functional activities of daily life that maintains mobility and independence in elderly people. The ability to precisely coordinate alternating limb movements for proper foot placement on each stair, high-level stability control for single limb balance during limb advancement, and proper lower extremity strength and joint range of motion are required for stair ambulation [26,27]. Stair climbing also requires approximately 30–40% more metabolic energy compared with level walking [28]. In elderly people, regular stair climbing may be a promising way to increase physical function, cardiovascular health, and muscle strength and maintain independence. However, the ability to participate in stair climbing is affected by age-related factors such as levels of physical activity, decreases in muscle strength and balance, increases in the energetic cost of walking, changes in visual acuity, multiple medications, cognitive decline, and lower extremity pain [29].

As they age, elderly people gradually lose their ability to walk and are at greater risk of injury compared with younger adults, particularly when walking uphill [30]. Inclined walking is associated with a higher metabolic rate, deviated gait patterns (including gait and stride time variability), and changes in the mediolateral center of mass and joint angles, all of which increase the risk of falls [31,32]. Incline treadmill walking is a popular form of exercise that is frequently used for training and rehabilitation. It provides multiple health benefits, including increased leg muscle activation and workload, improved postural balance, elevated heart rate, and higher metabolic rate, which can result in increased calorie consumption and burning of fat tissue compared with level walking [33]. However, age-related changes and mobility limitations can make it challenging for many elderly people to engage in inclined walking exercise.

A powered hip exoskeleton integrates robot power with human intelligence. The robotic system can train the wearer’s muscles and facilitate movement by providing controllable assistive and resistive force and torque at the hip joint. It also has the potential to improve locomotion, endurance, and strength in a variety of areas, including clinical rehabilitation, recreation, and demanding occupation-related tasks [34,35]. Compared with conventional exercise, robot-assisted exercise can facilitate highly controlled, repetitive, extensive, and task-specific exercise [36]. Wearable exoskeleton systems may represent a promising training tool for physical activity and exercise.

In this study, we used a wearable robotic hip exoskeleton, the EX1, which was developed at Samsung Electronics Co., Ltd. (Suwon, Republic of Korea), and compared the effects of different exercise interventions. The purpose of this study was to investigate the effects of EX1-assisted exercise in various environments on physical function, muscle strength, and walking efficiency in older adults.

## 2. Materials and Methods

### 2.1. Study Design and Participants

This study used a single-blinded (evaluator), randomized, controlled, four-group parallel design. All participants attended 12 experimental visits: 2 testing visits (pre- and post-test) and 10 exercise visits.

A total of 60 community-dwelling older adults were included in this study. We excluded individuals with a history of neurological disorders and musculoskeletal disorders that affect walking capacity, efficiency, and endurance. Eligible adults were healthy and between 65 and 85 years of age without a history of a central nervous system disease. Subject exclusion criteria were: (1) difficulty walking independently due to problems such as visual field defects or fractures, (2) severe arthritis or orthopedic problems that limit passive range of motion of the lower extremities, (3) severe cognitive decline (mini mental state examination—Korean ≤10) affecting the ability to fully understand the experimental procedure, (4) difficulty participating in exercise programs due to diseases such as uncontrolled hypertension and diabetes, (5) severe dizziness, and (6) obesity (defined as a body mass index greater than 35). The eligible participants were randomly assigned to group A, B, C, or D using a computer-generated 1:1:1:1 allocation.

The study procedures were approved by the ethics committee of Samsung Medical Center Institutional Review Board (Approval Number: 2021-03-052) and registered with ClinicalTrial.gov (NCT04920201). Written informed consent was obtained from all participants before they entered the study, and all methods were carried out in accordance with the approved study protocol.

### 2.2. Exercise

All participants were allocated to one of four groups: group A (overground walking without an EX1, *n* = 15), group B (overground walking using the resistance mode of an EX1, *n* = 15), group C (stair ascent using an EX1 in assistance mode, *n* = 15), and group D (inclined treadmill walking using an EX1 in assistance mode, *n* = 15).

The subjects in Group A performed overground walking exercise without using an EX1 at a comfortable speed for a straight distance of 300 m. The subjects in Group B performed overground walking exercise using the resistance mode of EX1 at a comfortable speed for a straight distance of 300 m with an average resistance torque of −4.6 Nm on the right and left sides. The subjects in Group C performed a stair ascent exercise using the assistance mode of EX1 with an average assistance torque of 8.2 Nm on the right and left sides. The participants climbed stairs from the first basement level to the fourth floor at a comfortable speed and then descended using the elevator before ascending again; this was performed a total of 4 times on average for 30 min. The subjects in Group D performed a walking exercise on a 16% (9.09°) incline treadmill using the assistance mode of EX1 at a comfortable speed with an average assistance torque of 8.3 Nm on the right and left sides.

The exercise interventions were conducted at a perceived exertion of between 12 and 16 (somewhat hard to hard) on Borg’s rating of perceived exertion scale [37] for 3 exercise sessions per week for 4 weeks (a total of 10 sessions). The duration of each exercise session was 40 min: 5 min of warm-up, 30 min of exercise, and 5 min of cool-down [38,39]. The exercise intensity was controlled by giving the subject a rest period and adjusting the assistive or resistive torque of the EX1 applied to the subject. Subjects were supervised by a physical therapist during intervention to maintain patient safety. If a participant missed an exercise session, an additional session was offered at another time of the week or at the end of the intervention period.

### 2.3. Assessment Tools and Data Collection

Participants were assessed at two time points: pre- and post-test. Each assessment evaluated physical function (performance-based and subject-reported measures), muscle strength, and cardiopulmonary metabolic energy efficiency. All assessments were conducted by an experienced physical therapist blinded to intervention assignments and collected when subjects were not wearing the EX1.

To measure physical function, the 10-meter walk test for self-selected velocity (10MWT-SSV), short physical performance battery (SPPB), Berg balance scale (BBS), timed up and go (TUG), and functional reach test (FRT) were performed. In addition, participants completed the geriatric depression scale—short form (GDS-SF) to monitor changes in mood status after exercise.

Muscle strength of the trunk and lower extremities was measured using a portable digital handheld dynamometer (MicroFET2, Hoggan Health Industries, Salt Lake City, UT, USA) designed specifically for taking objective, reliable, and quantifiable measurements of muscle performance. The averages of three maximal efforts of trunk flexion, trunk extension, hip flexion, hip extension, hip abduction, hip adduction, knee flexion, knee extension, ankle dorsiflexion, and ankle plantarflexion were evaluated for each individual. The muscle strength test at all time points was performed by the same physical therapist using the same measurement method.

Cardiopulmonary metabolic energy efficiency was measured using a validated and reliable portable telemetric gas analyzer system, K5 (COSMED, Rome, Italy). The COSMED K5 portable cardiopulmonary metabolic system used combined breath-by-breath technology to measure oxygen consumption and carbon dioxide production. The participants wore the system on their upper body with a face mask to prevent exposure to outside air for breath analysis. A facial mask was worn over the participant’s nose and mouth, with a gas sample line and analyzer unit strapped to the participant’s chest and connected to a battery-operated unit. To ensure proper operation of the K5 analyzer, the flow turbine and gas analyzer were calibrated using a 3-L calibration syringe, gas, and regulators prior to each test. Baseline cardiopulmonary metabolic energy efficiency was measured in a comfortable standing position for 5 min. Metabolic energy costs were then recorded during 6 min of treadmill walking at each participant’s most comfortable gait speed. Participants identified their most comfortable speeds by walking at their own pace for 3 min without using handrails. Metabolic energy cost tests at all time points were performed on the same treadmill at the same speed and using the same measurement method. The net cardiopulmonary metabolic cost (mL·kg^−1^·min^−1^) was calculated by subtracting the standing oxygen demand (baseline) from the average oxygen uptake during the last 2 min. The energy expenditure measurement (EEm) in Kcal/min was calculated in the same way.

### 2.4. EX1 Wearable Hip Exoskeleton

The EX1 is a hip-type robotic exoskeleton designed to improve ambulatory function in elderly people. The device is worn around the wearer’s waist and thighs to assist or resist hip joint flexion and extension. The EX1 consists of snap-together components weighing a total of 2.1 kg (Figure 1). The waist part houses two actuator modules, a Bluetooth module, and a control pack. Each actuator module includes a motor, an embedded angular position sensor, and a controller. The control pack contains a central processor, an inertial measurement unit sensor, a rechargeable battery pack, and a power switch. Assistive and resistive torque are exerted on the wearer’s thighs via thigh frames. The EX1 is controlled by a trained physical therapist who can change the assist and resist settings using software on a mobile device. The EX1 can be used for 2 h when walking continuously at 3 km/h. The maximum noise level is less than 60 dB at a distance of 1 m.

The EX1 has one operating mode: delayed output feedback control (DOFC). This time-delayed, self-excited feedback control method does not include a gait phase estimator or a reference lookup for generating assistive torque. Assistive torque is immediately applied following movement of the user by updating the change in hip motion at every control period (100 Hz). In DOFC mode, an onboard microprocessor receives signals from integrated sensors that provide information on the wearer’s joint angles. Sensors in the device detect the hip joint angle and associated angular velocity, based on which, the central processor determines the optimal assistive or resistive torque output. The actuator module then exerts the assistive or resistive force on the user’s hip joint to reduce or increase the physical effort in walking, which in turn motivates walking and stimulates weak areas in the muscular system related to walking [40].

The EX1 defines the angle of the hip joint to be 0° when the user is wearing the device and standing upright. It has mechanical stoppers that are set to a maximum of 110° for flexion and a maximum of 45° for extension. The mechanical stoppers are designed to ensure that the hip joint stays within the specified range even in the case of device malfunction. In addition, the DOFC controller in normal use is set to a maximum of 100° for flexion and a maximum of 40° for extension. Furthermore, the device alerts the physical therapist by sending out a warning message via the mobile application when the joint angle reaches the joint angle limit. If clothes become caught in the device, the torque-off function is automatically activated to reduce risk, at which point the device does not provide assistive or resistive force. The working torque range provided by the EX1 is 0–12 Nm (−12-0 Nm for resistance torque), and the maximum torque is limited to 12 Nm + 15%. The device is designed to stop at any time when there is user intention to stop. In other words, the device stops assistive or resistive force (torque off) when the device judges that the user has stopped walking. Thereafter, when the physical therapist clears the situation and the user starts to walk again, the device starts to provide assistive or resistive force to the user.

### 2.5. Statistical Analysis

Statistical analyses were performed using SPSS version 22.0 (IBM, Armonk, NY, USA), and the significance level was set at 0.05. Descriptive statistics are expressed as the mean (standard deviation (SD)). To determine the appropriate statistical tests to apply, we checked the distribution of the data for normality, and, consequently, we applied parametric tests for 10MWT-SSV, TUG, FRT, muscle strength of trunk flexion, trunk extension, hip flexion, hip extension, hip abduction, hip adduction, knee flexion, knee extension, ankle dorsiflexion, net metabolic energy cost, and EEm and non-parametric tests for SPPB, BBS, GDS-SF, and muscle strength of plantar flexion. One-way analysis of variance (ANOVA) and Kruskal–Wallis tests for continuous variables and chi-square tests for categorical variables were used to compare participants’ baseline characteristics. To evaluate intervention effects within groups, paired *t*-tests and Wilcoxon signed-rank tests were used to compare the outcome measures before and after EX1 exercise in each group. One-way ANOVA and Kruskal–Wallis tests were used to determine statistically significant differences among groups. In addition, repeated-measures ANOVA was used to examine the main effects of exercise over time, including groups and time points. Post hoc tests were used to determine whether there were differences among the group means, and the significance levels of the tests were adjusted using Bonferroni corrections.

## 3. Results

### 3.1. Subjects

A total of 67 community-dwelling older adults were enrolled in screening, and seven subjects who did not meet the inclusion criteria (*n* = 5) or declined to participate (*n* = 2) were excluded. Sixty eligible participants were assigned using simple randomization procedures to Groups A, B, C, or D and received the allocated intervention. The final analysis included 58 subjects after excluding two subjects in Group A who discontinued the intervention (Table 1). The experimental groups did not differ in general characteristics, confirming that baseline conditions were similar among groups.

### 3.2. Effect of EX1 Exercise on Physical Function

Changes in physical function at the pre and post time points in each group are shown in Figure 2. A significant increase in 10MWT-SSV and TUG test scores was observed in groups B, C, and D (*p* < 0.05 and *p* < 0.01, respectively), but not in group A. For the SPPB and FRT, a statistically significant improvement was evident only in group D (*p* < 0.05 and *p* < 0.01, respectively). BBS showed significant improvement in all groups (*p* < 0.01), and GDS-SF scores decreased significantly after exercise with an EX1 in groups B and D (*p* < 0.05). There were no statistically significant interactions between group and time for physical function.

The pre–post difference values and the minimal clinically important difference (MCID) in physical function are presented in Table 2. Groups B, C, and D showed significantly greater changes in 10MWT-SSV compared with the MCID, and group D exhibited a significantly greater change in SPPB than in MCID. However, no statistically significant differences were seen in pre–post changes among groups with respect to physical function.

### 3.3. Effect of EX1 Exercise on Muscle Strength

Changes in muscle strength of the trunk and lower extremities at the pre and post time points are shown in Table 3. Group A showed significant changes in muscle strength of knee flexion and extension after overground walking without an EX1 (*p* < 0.05 and *p* < 0.01, respectively), while group B showed significant improvements in the muscle strength required for trunk flexion, hip flexion, hip extension, hip abduction, knee flexion, knee extension, ankle dorsiflexion, and ankle plantar flexion after overground walking using the EX1′s resistance mode (*p* < 0.05, *p* < 0.01). For group C, significant improvements were seen in the muscle strength of trunk flexion, hip adduction, knee extension, and ankle plantarflexion after stair ascent exercise using the EX1′s assistance mode (*p* < 0.05, *p* < 0.01). Group D showed significant changes in the muscle strength of trunk extension, hip flexion, hip extension, and knee extension after inclined treadmill walking using the assistance mode of the EX1 (*p* < 0.05, *p* < 0.01).

### 3.4. Effect of EX1 Exercise on Cardiopulmonary Metabolic Energy Efficiency

Values at the pre and post time points for net metabolic energy cost (mL·kg^−1^·min^−1^) and EEm (Kcal/min) are shown in Figure 3. Group C showed a significant change in net metabolic energy cost (a reduction of 12.80%) and EEm (reduction of 10.03%) after stair ascent exercise using the EX1 in assistance mode (*p* < 0.05). Group D also showed significant changes in net metabolic energy cost (a reduction of 21.66%) and EEm (a reduction of 18.30%) after inclined treadmill walking using the EX1 in assistance mode (*p* < 0.05). No significant differences were observed in groups A and B.

## 4. Discussion

The purpose of this study was to investigate the effects of exercise with EX1 in various environments on physical function, muscle strength, and walking efficiency in older adults. The findings of this study suggest that exercise using an EX1 in various environments offers several key advantages over exercise without an EX1 in terms of physical function, muscle strength, and cardiopulmonary metabolic energy efficiency in older people. Physical function (10MWT-SSV, SPPB, TUG, FRT, and GDS-SF) improved more after exercise with an EX1 than without an EX1, and muscle strength also increased more in the exercise groups that used an EX1 than those who did not use an EX1, particularly in Group B, in which exercises were performed using the resistance mode of EX1. Furthermore, the net metabolic energy costs and EEm during walking significantly improved in exercise groups C and D, in which exercises were performed in assistance mode.

Within the natural processes of aging, decreases in gait speed are common [41,42]. Slowing of gait with aging appears to be a universal biological phenomenon and likely reflects the integrated performance of numerous organ systems. Factors affecting gait ability can be classified into six main physiological subsystems: the central nervous system, peripheral nervous system, perceptual system, muscles, bone and/or joints, and energy production and/or delivery. When any of these systems becomes dysfunctional as a result of frailty-associated progressive decline, walking can slow [43,44,45]. Gait speed has been recommended as a potentially useful clinical indicator of well-being among older adults and may also be a simple and accessible summary indicator of vitality because it integrates known and unrecognized impairments in multiple organ systems that affect survival [46,47]. In this study, 10MWT-SSV was evaluated before and after exercise intervention to investigate the effect of EX1 exercise on gait speed. The results showed that gait speed improved significantly after EX1 exercise in groups B, C, and D, but not in group A. In addition, only groups B, C, and D, in which exercise was performed with the help of an EX1, showed significantly greater changes in 10MWT-SSV compared with the MCID, which refers to the smallest change in an outcome that represents a meaningful health change for the elderly. The EX1 exerts assistive or resistive force on the user’s hip joint to reduce or increase the physical effort involved in walking, which in turn motivates walking and stimulates weak areas in the muscular system related to walking. Furthermore, the EX1 can provide effective help for repetitive and intensive gait-training interventions and allow a freer and more natural movement while walking, which can have a positive effect on the improvement of gait speed in the elderly. Thus, decreased gait speed common in older adults could be improved through EX1 exercise.

Balance disorder is common in elderly people and is a major cause of falls in this population. Falling is associated with reduced physical function, limited quality of life, and a loss of independence, as well as increased morbidity and mortality [6,48,49]. Because most balance disorders in older persons are multifactorial in origin, an appropriate physical activity and exercise program is recommended to restore, maintain, or improve functional abilities [50]. A variety of exercise interventions, including walking, functional exercise, muscle strengthening, and combinations of the different exercises, have been found to significantly improve balance [51]. In this study, only groups B, C, and D, who performed exercise with the EX1, showed significant improvements in TUG tests. Only group D, who performed inclined treadmill walking exercise using an EX1 in assistance mode, showed significant improvements in SPPB and FRT. These results indicate that exercise using different EX1 modes (assistance and resistance) in various environments (overground, stair, and inclined walking) could improve gait and balance, and EX1 exercise can be recommended as an exercise program to help improve age-related changes in gait and balance. In particular, group D, who performed inclined treadmill walking exercise using the assistance mode of EX1, showed significant improvement in all physical functions. Compared with level ground walking, the ability to walk on an incline requires a different lower extremity motor pattern. This pattern requires increased force output by the lower extremity muscles and increased range of motion, stimulating the lower extremity muscles [52,53]. Using a treadmill to exercise instead of walking outside can reduce the risk of injury to the hip, knee, and ankle joints because an inclined treadmill increases intensity levels without stressing the body’s joints. Previous studies have indicated that gait training on an inclined treadmill can be helpful for maintaining muscle strength and heart health [54,55,56]. The results of this study suggest that intensive inclined treadmill exercise with an EX1 may lead to improved mobility and balance in older adults.

In addition, participants completed the GDS-SF to monitor changes in mood status after exercise. Previous studies have shown that improvements in physical function are generally related to a lower incidence of depressive symptoms and improved mood status among older adults. Physical improvements are also associated with superior mental health, well-being, and quality of life [57,58]. In this study, members of groups B and D, who showed great improvement in muscle strength and physical function after exercise with EX1, showed a significant change in GDS-SF scores. Improved physical functioning after EX1-assisted exercise may therefore have a positive effect on mood status among elderly people.

Older adults generally have muscle weakness [59]. Muscle weakness is the main factor in dysfunction of locomotor activity and balance in elderly people. One of the reasons for the development of muscle weakness in elderly people is decreased physical activity [60]. As a physiological process, aging involves a gradual decrease in skeletal muscle endurance, and this is related to a reduction in fitness. A decrease in physical activity and exercise provides a theoretical background for the use of both endurance and resistance exercise as interventions to improve health outcomes in elderly people [61]. In a previous study, elderly people were 59% weaker compared with young subjects, but 6 months of resistance training improved muscle strength in members of the older group, who were then only 38% weaker than the younger group [62]. In this study, muscle strength of the trunk and lower extremities increased more in the exercise groups that used EX1 than those without EX1, particularly in Group B, which performed overground walking exercises using the EX1 in resistance mode. Furthermore, the EX1, which applies torque only to the hip joint, affects not only the strength of the hip joint muscles, but also the strength of the trunk and ankle muscles. When walking is performed using the resistance mode of the EX1, it is possible to experience both aerobic and resistance effects at the same time, and a more comprehensive and practical effect on the physical performance of elderly people is often evident. Endurance exercise induces improvements in VO2max and submaximal endurance capacity in the elderly. Strength exercise is therefore an effective intervention for improving muscle strength, power output, and muscle mass in these populations. A combination of endurance and strength exercises with EX1 appears to be the most effective strategy for improving neuromuscular function [63]. In other words, the EX1 can have a positive effect on muscle strength when it is used as a training tool during endurance, resistance, and multicomponent exercises (combined strength and endurance exercises) in elderly people.

A constellation of age-related walking problems, including slow walking, poor stability, and uncoordinated timing of stepping with gait postures and phases, contributes to inefficient gait [64,65,66,67]. The loss of motor skills and the related high energy cost of walking (i.e., walking inefficiently) are major factors in the age-related decline in physical function and activity for older adults [64]. In this study, groups C and D, who climbed stairs and performed inclined treadmill walking exercise in EX1 assistance mode, showed significant changes in net metabolic energy cost and EEm after EX1 exercise. Stair climbing and inclined walking exercise provide ubiquitous and cost-effective opportunities to incorporate physical exercise into daily routines [68]. Undertaking regular bouts of physically demanding exercise is beneficial for general health [69]. Indeed, stair climbing and inclined walking have been shown to enhance muscle recruitment, improve cardiovascular capacity, increase energy expenditure, and improve calorie burn [70,71]. However, the ability to participate in stair climbing and inclined walking is affected by decreased levels of physical activity and age-related changes. The EX1 allows the elderly to participate in stair and inclined walking exercises by providing assistance torque. It also enables a more free and natural movement while walking with consistent and high-dose repetition of movement, which could lead to a reduction in cardiopulmonary metabolic costs.

This study has some limitations. First, the statistical power was low because of the small number of participants. Second, the intervention and follow-up periods were relatively short. Therefore, this result cannot be generalized to the entire elderly population, and long-term training with long-term follow-up of the training effects in larger participant groups needs to be conducted in the future. In spite of these limitations, this study demonstrated the superior effects of the EX1 for assisting with overground, stair, and inclined treadmill walking, leading to greater improvements in gait efficiency and physical function, compared to exercise without EX1. In addition, all participants completed experimental protocols without any specific adverse events, indicating that EX1 exercise is safe and does not pose risks for older adults. Nonetheless, several participants suggested improvements to the weight, noise, and design of EX1. Future works will examine the usability of and satisfaction with the EX1.

This study was a randomized controlled trial evaluating physical function, muscle strength, and walking efficiency in elderly people after exercise with an EX1. These findings support application of the EX1 to physical activity and exercise to improve age-related declines in physical function and walking efficiency. In addition, personalized exercise programs using different modes and training environments for the EX1 can be applied to enhance physical function and walking efficiency of the elderly.

## 5. Conclusions

Physical activity and exercise are recommended strategies to mitigate age-related declines in gait and physical function in older adults. The clinical significance of exercise intervention is worthy of further study. The findings in this study support application of the EX1 to physical activity and exercise to improve age-related changes in physical function, muscle strength, and walking efficiency among older adults. In addition, personalized exercise programs using different modes and training environments with the EX1 can enhance physical function and walking efficiency among the elderly.

## Figures and Tables

**Figure 1 jpm-12-02077-f001:**
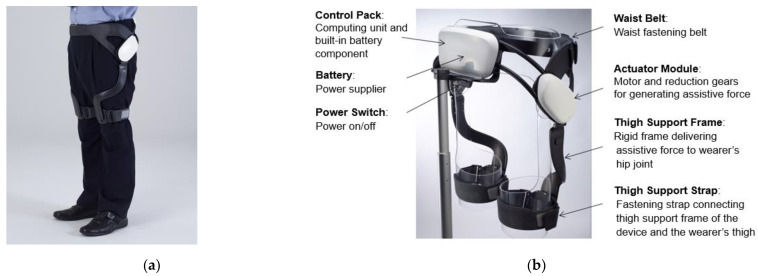
Wearable Hip Exoskeleton, EX1: (**a**) Exterior of EX1; (**b**) Description of the exterior.

**Figure 2 jpm-12-02077-f002:**
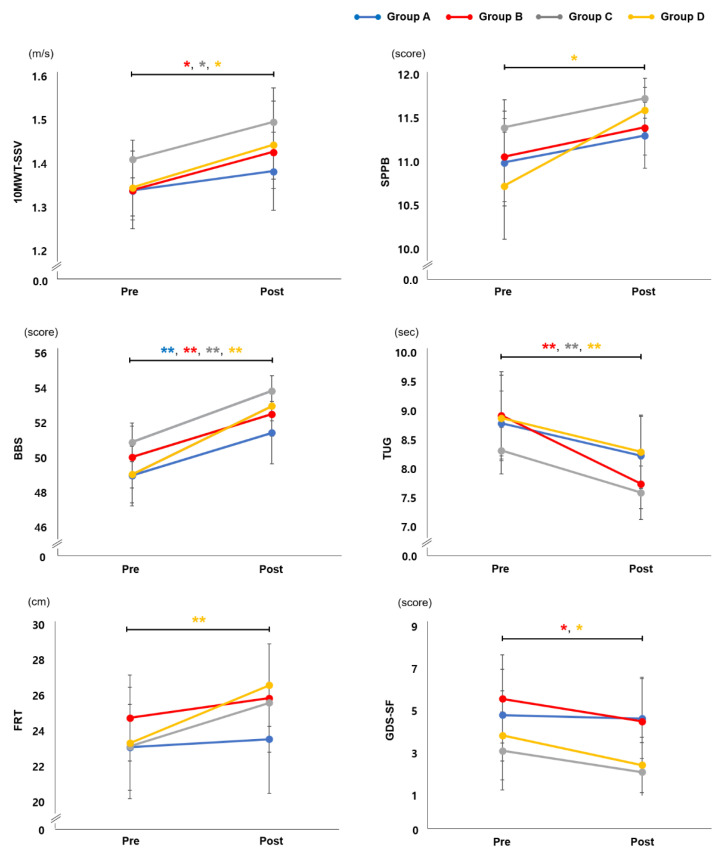
Physical Function. Changes in physical function at the pre and post time points in groups A, B, C, and D. Group A: overground walking without an EX1, group B: overground walking using the resistance mode of EX1, group C: stair ascent using the assistance mode of EX1, group D: inclined treadmill walking using the assistance mode of EX1. Paired *t*-tests were used for 10MWT-SSV, TUG, and FRT and Wilcoxon signed-rank tests for SPPB, BBS, and GDS-SF. * *p* < 0.05 compared with pre, ** *p* < 0.01 compared with pre. 10MWT-SSV, 10-Meter Walk Test for self-selected velocity; BBS, Berg Balance Scale; FRT, Functional Reach Test; GDS-SF, Geriatric Depression Scale—Short Form; SPPB, Short Physical Performance Battery; TUG, Timed Up and Go.

**Figure 3 jpm-12-02077-f003:**
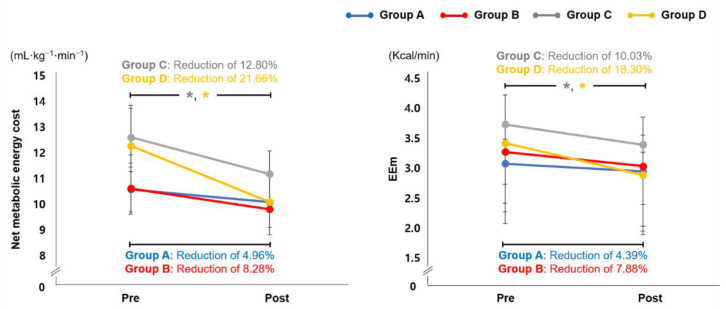
Cardiopulmonary Metabolic Energy Cost. Changes in cardiopulmonary metabolic energy cost at the pre and post time points in groups A, B, C, and D. Group A: overground walking without an EX1, group B: overground walking using the resistance mode of EX1, group C: stair ascent using the assistance mode of EX1, group D: inclined treadmill walking using the assistance mode of EX1. * *p* < 0.05 compared with pre, paired *t*-test. EEm, energy expenditure measurement.

**Table 1 jpm-12-02077-t001:** Baseline Characteristics of Participants (*N* = 58).

Characteristics	Group A	Group B	Group C	Group D	χ2/F(*p*)
Sex (male/female)	6/7	8/7	9/6	7/8	0.742 (0.863)
Age, years	76.38 (4.98)	75.20 (3.41)	72.73 (3.10)	74.67 (5.05)	1.866 (0.146)
Height, cm	157.62 (9.90)	163.90 (5.86)	162.87 (9.89)	160.53 (6.97)	1.571 (0.207)
Weight, kg	59.50 (9.92)	62.80 (8.84)	62.03 (12.88)	59.40 (10.81)	0.382 (0.766)
BMI, kg/m^2^	24.01 (3.89)	23.43 (3.57)	23.22 (3.43)	22.93 (2.94)	0.242 (0.867)
MMSE-K	25.92 (2.06)	27.27 (1.44)	27.80 (2.21)	27.07 (2.02)	6.214 (0.102)

Data are expressed as the mean (standard deviation). Group A: overground walking without an EX1, group B: overground walking using the resistance mode of EX1, group C: stair ascent using the assistance mode of EX1, group D: inclined treadmill walking using the assistance mode of EX1. BMI, body mass index; MMSE-K, mini mental state examination—Korean.

**Table 2 jpm-12-02077-t002:** Pre–Post Change and MCID in Physical Function and Depression (*N* = 58).

	Group A	Group B	Group C	Group D	χ2/F(*p*)
Δ10MWT-SSV, m/s	0.04 (0.17)	0.09 (0.13) ^a^	0.09 (0.12) ^a^	0.10 (0.15) ^a^	0.381 (0.767)
ΔSPPB	0.31 (0.63)	0.33 (0.82)	0.33 (0.62)	0.87 (1.19) ^a^	2.514 (0.473)
ΔBBS	2.46 (2.03)	2.46 (1.69)	2.95 (2.47)	3.93 (2.15)	4.539 (0.209)
ΔTUG, sec	0.56 (1.05)	1.18 (0.95)	0.73 (0.81)	0.58 (0.61)	1.631 (0.193)
ΔFRT, cm	0.46 (2.56)	1.12 (5.47)	2.45 (6.13)	3.38 (4.18)	1.048 (0.379)
ΔGDS-SF	0.16 (1.77)	1.07 (1.49)	1.00 (2.18)	1.40 (2.03)	4.109 (0.250)

Data are expressed as the mean (standard deviation). Group A: overground walking without an EX1, group B: overground walking using the resistance mode of EX1, group C: stair ascent using the assistance mode of EX1, group D: inclined treadmill walking using the assistance mode of EX1. ^a^ Significantly greater than the minimal clinically important difference: 10MWT-SSV = 0.05 m/s, SPPB = 0.5 for older adults. 10MWT-SSV, 10-Meter Walk Test for self-selected velocity; BBS, Berg Balance Scale; FRT, Functional Reach Test; GDS-SF, Geriatric Depression Scale-Short Form; MCID, Minimal Clinically Important Difference; SPPB, Short Physical Performance Battery; TUG, Timed Up and Go.

**Table 3 jpm-12-02077-t003:** Effect of EX1 Exercise on Muscle Strength (*N* = 58).

Muscle Strength (kg)	Group A		Group B		Group C		Group D	
	Pre	Post	Pre	Post	Pre	Post	Pre	Post
Trunk flexion	21.62 (6.98)	22.84 (5.55)	18.84 (5.08)	24.19 (6.66) **	24.16 (5.08)	27.77 (7.29) *	21.94 (6.33)	24.06 (5.90)
Trunk extension	26.26 (6.58)	27.47 (9.61)	24.78 (6.75)	26.52 (4.54)	28.20 (6.99)	30.28 (9.86)	26.81 (7.40)	31.92 (5.43) **
Hip flexion	25.94 (8.35)	28.63 (9.17)	28.80 (6.52)	33.48 (6.71) *	31.12 (8.16)	32.92 (7.45)	28.82 (8.92)	33.36 (10.12) *
Hip extension	16.64 (7.28)	19.35 (6.20)	17.82 (6.46)	21.70 (6.42) **	20.44 (5.54)	22.99 (6.31)	18.22 (6.61)	22.90 (8.15) **
Hip abduction	18.89 (6.81)	21.17 (7.01)	18.15 (4.96)	24.63 (5.22) **	24.11 (6.30)	25.55 (7.02)	20.37 (6.56)	22.90 (8.38)
Hip adduction	34.12 (13.29)	34.59 (10.09)	36.65 (11.74)	38.38 (8.45)	34.00 (9.44)	41.44 (9.12) **	36.92 (12.34)	38.90 (12.36)
Knee flexion	30.68 (13.11)	34.87 (9.82) *	35.60 (10.66)	40.10 (9.21) *	40.08 (9.92)	41.98 (11.64)	39.99 (12.19)	44.59 (13.31)
Knee extension	28.89 (7.62)	33.77 (4.90) **	32.12 (8.29)	35.38 (8.73) **	34.08 (4.45)	37.38 (5.92) *	30.70 (5.14)	35.66 (8.17) **
Ankle DF	32.87 (10.88)	34.58 (12.87)	36.16 (10.52)	42.29 (9.67) *	39.70 (9.40)	44.32 (10.79)	38.37 (10.60)	40.06 (11.97)
Ankle PF	38.26 (13.83)	42.32 (14.81)	31.39 (10.16)	43.11 (14.71) ^§§^	36.15 (8.48)	44.30 (15.34) ^§^	42.48 (12.60)	52.62 (19.11)

Data are expressed as the mean (standard deviation). Group A: overground walking without an EX1, group B: overground walking using the resistance mode of EX1, group C: stair ascent using the assistance mode of EX1, group D: inclined treadmill walking using the assistance mode of EX1. * Significant change compared with pre (paired *t*-test, *p* < 0.05), ** Significant change compared with pre (paired *t*-test, *p* < 0.01). ^§^ Significant change compared with pre (Wilcoxon signed-rank test, *p* < 0.05), ^§§^ Significant change compared with pre (Wilcoxon signed-rank test, *p* < 0.01). DF, dorsiflexion; PF, plantarflexion.

## Data Availability

Data are available from the corresponding author upon request.

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
