# Peer review of "Effect of Exercise Using an Exoskeletal Hip-Assist Robot on Physical Function and Walking Efficiency in Older Adults"

_jpm, 2022, doi:10.3390/jpm12122077_

Round 1
Reviewer 1 Report
The study entitled "Effect of Exercise Using an Exoskeletal Hip-assist Robot on Physical Function and Walking Efficiency in Older Adults" was interesting and well conducted, however, the introduction should be concise and polished. The limited number of subjects affects the conclusion of the study and we cannot use these results for the general population as the authors mentioned in the limitations of the study. On the other hand, it would be better if the authors followed up with the patients for a longer period and also asked them about their satisfaction with the device and also the improvement of their quality of life. It is obvious that this device could improve the performance of elderly patients, but something that is more important is how to use it routinely every day. I think the authors could add this part to their limitations of the study.
Author Response
We agree with the reviewer’s comments and added limitations in the Discussion section of the revised manuscript as follows.
“This study has some limitations. First, the statistical power was low because of the small number of participants. Second, the intervention and follow-up periods were relatively short. Therefore, this result cannot be generalized to the entire elderly population and long-term training with long-term follow-up of the training effects in larger participant groups needs to be conducted in the future. In spite of these limitations, this study demonstrated the superior effects of the EX1 for assisting with overground, stair, and inclined treadmill walking, leading to greater improvements in gait efficiency and physical function, compared to exercise without EX1. In addition, all participants com-pleted experimental protocols without any specific adverse events, indicating that EX1 exercise is safe and does not pose risks for older adults. Nonetheless, several participants suggested improvements to the weight, noise, and design of EX1. Future works will examine the usability of and satisfaction with the EX1.”
Reviewer 2 Report
Brief summary: The present work investigated the effects that EX1-assisted exercise in various environments has on muscle strength, physical function, and walking efficiency of older adults. Results demonstrated that the application of the EX1 to physical activity and exercise can improve age-related changes in these factors among older adults. In addition, the assistance of the EX1 in personalized exercise programs using different modes and training environments can enhance physical performance and walking efficiency in elderly people.
Major comments:
The topic of this work is original, causing a probable interest in the reader. The introduction is clear and well written. Moreover, it refers to appropriate and recent literature studies. However, which gaps your work is filling? It should be explained to justify the development of the study.
Even if materials and methods are clearly explained, some improvements could be made. First, exclusion criteria for participants can be expressed through a bulleted list for a better clarity. Then, considering the statistical tests, why do you apply both a parametric and a non-parametric analysis? Did you verify the normal distribution of data to decide which test to apply?
Results are significant, clearly presented, and deeply discussed.
English should be checked considering both grammar and syntax.
Minor comments:
- Line 18: replace “investigate” with “investigated”
- Line 18: define the acronym EX1
- Line 58: replace “can help improve or maintain” with “can help improving or maintaining”. Moreover, replace “reduce” with “reducing”
- Line 68: replace “to improving” with “to improve”
Author Response
Major Comments:
- The topic of this work is original, causing a probable interest in the reader. The introduction is clear and well written. Moreover, it refers to appropriate and recent literature studies. However, which gaps your work is filling? It should be explained to justify the development of the study. Even if materials and methods are clearly explained, some improvements could be made. First, exclusion criteria for participants can be expressed through a bulleted list for a better clarity.
Response: As the reviewer pointed out, we revised and supplementary material and methods and results sections to provide clarity.
(1) In line 125 of the revised manuscript, we added a detailed description of the subject exclusion criteria as follows.
“Subject exclusion criteria were: 1) difficulty walking independently due to problems such as visual field defects or fractures, 2) severe arthritis or orthopedic problems that limit passive range of motion of the lower extremities, 3) severe cognitive decline (Mini Mental State Examination-Korean ≤ 10) affecting the ability to fully understand the experimental procedure, 4) difficulty participating in exercise programs due to diseases such as uncontrolled hypertension and diabetes, 5) severe dizziness, and 6) obesity (defined as a body mass index greater than 35).”
(2) In line 143 of the revised manuscript, we added a detailed description for the exercise as follows.
“The subjects in Group A performed overground walking exercise without using an EX1 at a comfortable speed for a straight distance of 300 m. The subjects in Group B performed overground walking exercise using the resistance mode of EX1 at a comfortable speed for a straight distance of 300 m with an average resistance torque of -4.6 Nm on the right and left sides. The subjects in Group C performed a stair ascent exercise using the assistance mode of EX1 with an average assistance torque of 8.2 Nm on the right and left sides. The participants climbed stairs from the first basement level to the fourth floor at a comfortable speed and then descended using the elevator before ascending again; this was performed a total of 4 times on average for 30 minutes. The subjects in Group D performed a walking exercise on a 16% (9.09°) incline treadmill using the assistance mode of EX1 at a comfortable speed with an average assistance torque of 8.3 Nm on the right and left sides.”
(3) In line 224 of the revised manuscript, we added a detailed description for the EX1 Wearable Hip Exoskeleton criteria as follows.
“The EX1 defines the angle of the hip joint to be 0° when the user is wearing the device and standing upright. It has mechanical stoppers that are set to a maximum of 110° for flexion and a maximum of 45° for extension. The mechanical stoppers are de-signed to ensure that the hip joint stays within the specified range even in the case of device malfunction. In addition, the DOFC controller in normal use is set to a maximum of 100° for flexion and a maximum of 40° for extension. Furthermore, the device alerts the physical therapist by sending out a warning message via the mobile application when the joint angle reaches the joint angle limit. If clothes become caught in the device, the torque-off function is automatically activated to reduce risk, at which point the de-vice does not provide assistive or resistive force. The working torque range provided by the EX1 is 0-12 Nm (-12-0 Nm for resistance torque), and the maximum torque is limited to 12 Nm+15%. The device is designed to stop at any time when there is user intention to stop. In other words, the device stops assistive or resistive force (torque off) when the device judges that the user has stopped walking. Thereafter, when the physical therapist clears the situation and the user starts to walk again, the device starts to provide assistive or resistive force to the user.”
(4) In line 268 of the revised manuscript, we added a detailed description of the subjects as follows.
“A total of 67 community-dwelling older adults were enrolled in screening, and seven subjects who did not meet the inclusion criteria (n=5) or declined to participate (n=2) were excluded. Sixty eligible participants were assigned using simple randomization procedures to Groups A, B, C, or D and received the allocated intervention. The final analysis included 58 subjects after excluding two subjects in Group A who dis-continued the intervention (Table 1).”
- Considering the statistical tests, why do you apply both a parametric and a non-parametric analysis? Did you verify the normal distribution of data to decide which test to apply?
Response: We added a detailed description of the statistical tests as follows.
“To determine the appropriate statistical tests to apply, we checked the distribution of the data for normality, and consequently we applied parametric tests for 10MWT-SSV, TUG, FRT, muscle strength of trunk flexion, trunk extension, hip flexion, hip extension, hip abduction hip adduction, knee flexion, knee extension, ankle dorsiflexion, net metabolic energy cost, and EEm and non-parametric tests for SPPB, BBS, GDS-SF, and muscle strength of plantar flexion.”
- Results are significant, clearly presented, and deeply discussed. English should be checked considering both grammar and syntax.
Response: English language editing was performed by a commercial English-language editing service (eWorld Editing, Eugene, OR, USA).
Minor Comments:
- Line 18: replace “investigate” with “investigated”
Response: We replaced “investigate” with “investigated”.
- Line 18: define the acronym EX1
Response: Actually, EX1 is the research project name, not an abbreviation for something.
- Line 58: replace “can help improve or maintain” with “can help improving or maintaining”. Moreover, replace “reduce” with “reducing”
Response: We replaced “can help improve or maintain physical function and overall health and reduce~” with “can help with improving or maintaining physical function and overall health and reducing~”.
- Line 68: replace “to improving” with “to improve”
Response: We replaced “to improving” with “to improve”.
Round 2
Reviewer 1 Report
Accept